# Updating an allocentric goal from lateralised egocentric visual memories

Antoine Wystrach [1] ✉, Florent Le Moël [1,2], Leo Clement[1] & Sebastian Schwarz[1,3]

Animals navigate by combining egocentric (viewpoint-dependent) and allocentric (world-referenced) spatial representations, yet how their brains achieve this integration remains unclear. Here we show how the brains of insect expert navigators, such as ants, accomplish this task. Field experiments reveal that ants recognise long-term egocentric visual memories – assumed to be encoded in the Mushroom Bodies – via a lateralized mechanism: instead of memorising views while facing their goal, ants store these memories by looking to the sides. Recognition signals inform whether to turn left or right, but do not directly drive motor responses. Instead, they are processed separately – presumably in the two brain hemispheres – and integrated to update a goal heading in an ancestral, central brain region –the central complex. This goal heading —now anchored in an allocentric frame—is then used with celestial compass cues for robust steering. Computational models based on insect neural circuits validate this two-stage process, demonstrating how noisy, viewpoint-dependent lateralized inputs are transformed into stable allocentric directional control. These findings reveal how compact brains leverage bilateral processing to combine spatial representations for visual navigation.

It is long known that insect expert navigators such as ants, wasps and bees store long-term egocentric visual memories of their surroundings[1,2], which are encoded in a brain area called the Mushroom Bodies (MB)[3–6]. These visual memories are stored in an egocentric way, requiring the insects—despite their nearly full-panoramic visual field[7–9]—to align their gaze direction with what they experienced previously in order to recognise a view[5,10–12]. For instance, ants can no longer recognise the natural scene along a familiar one-way route if their gaze is oriented away from the usual direction, such as when dragging a heavy food item backwards[13]. This calls into question their ability to perform mental rotations of the perceived scene and explains why ants, wasps, and bees need to physically 'scan' multiple directions to recognise memorised views and guide their path accordingly[13–17].

The observation that these insects regularly 'turn back and look' while departing from their nest or a feeder led to the dominant idea that they memorise egocentric views while facing the goal[13,17–23] (or

anti-goal[24–26]), and must consequently align their body in these same directions to recognise a learnt view as familiar when subsequently returning to the goal[12,20,25,27–33]. This is parsimonious as view familiarity —because it is rotation dependant—indicates that the goal is ahead and can simply trigger a 'go forward' command[30,34]. However, it implies that recognition bears no indication as to whether to turn left or right[12,20,25,27–33].

It should be noted that ants and flies, when subjected to a sudden rotation of the surrounding visual scene—in the absence of celestial compass cues—are able to correct their course accordingly[35,36]. This ability can be attributed to a short-term mapping between local visual features and their heading representation in another, ancestral brain area: the central complex (CX)[35,37–39]. This mapping does not involve the MB long-term visual memories, which in contrast to the CX, enables—and are necessary for[3,4]—the insect to recognise its familiar route after a large displacement.

[1]Centre de Recherches sur la Cognition Animale, Centre de Biologie Intégrative, Université de Toulouse, CNRS, Toulouse, France. [2]School of Informatics, University of Edinburgh, Edinburgh, United Kingdom. [3]Department of Biology, University of Graz, Graz, Austria. ✉e-mail: antoine.wystrach@utoulouse.fr

But how are these long-term, egocentric memories used for steering? Are they learned only when the animal is facing the goal or anti-goal direction? Does their recognition convey information about whether the goal lies to the left or right? Are these mushroom body (MB) visual recognition signals transferred into an allocentric representation in the CX? And if so, how is this transformation achieved?

We investigated these questions by combining neural modelling with field experiments in *Cataglyphis velox* and *Myrmecia croslandi* ants−two ecologically and phylogenetically distant species that rely on long-term visual memories for route navigation[40–43].

## Result

### The recognition of familiar views triggers compensatory left or right turns

We first tested whether ants could recognise views misaligned with their goal direction using an open-loop trackball system enabling the experimenter to choose both the position and body orientation of tethered ants directly in their natural environment[44]. We captured homing individuals of *C. velox* and *M. croslandi* species at the end of their two-way familiar route, just before they entered their nest. This ensured that these so-called zero-vector ants (ZV) can no longer rely on their path integration homing vector and must therefore rely on their terrestrial cue memories to recover their route direction[45]. We recorded the motor response of these ants while mounted on the trackball system, placed in the middle of their familiar route, far from the catchment area of the nest, when fixed in eight different body orientations (Fig. 1a, b). Results show that, irrespective of their body orientation, ants steered towards the correct route direction (Fig. 1,

Supplementary Fig. 1). This was not the case when tested in unfamiliar surroundings (Fig. 1c, d), showing that the lateralised responses observed on the familiar route were triggered by the recognition of the terrestrial visual scene. This demonstrates that ants can recognise familiar views of their route in multiple gaze orientations, and can derive whether the route direction is towards their left or their right. Interestingly, when the body was oriented towards (0°) or away (180°) from the nest/homing route direction, individuals still showed a strong preference for turning on one side (to the left or to the right, depending on individuals) (Fig. 1d). Perhaps it is possible to identify, for each individual, a precise orientation that triggers purely forward movement. However, we note that even when ants are free to recapitulate their familiar route on the ground, they continuously alternate between left and right turns rather than proceed straight[40,46–49]. Thus, even when facing the route or anti-route direction, recognition of familiar views appears to trigger a 'left vs. right' decision rather than a 'go forward vs. turn' decision.

### Guidance based on memorised views involves the celestial compass

We showed that the recognition of long-term visual memories along their route−known to be achieved in the MB[3–6]−indicates whether the goal direction is towards the left or right. In principle, guidance could thus be achieved by having these left/right signals directly trigger the left or right motor command. An alternative would be that the visual recognition signals from the MB are used to update a 'desired heading direction' in the CX, which in turn uses its own compass information to control steering[50,51]. This, combined with what we have shown

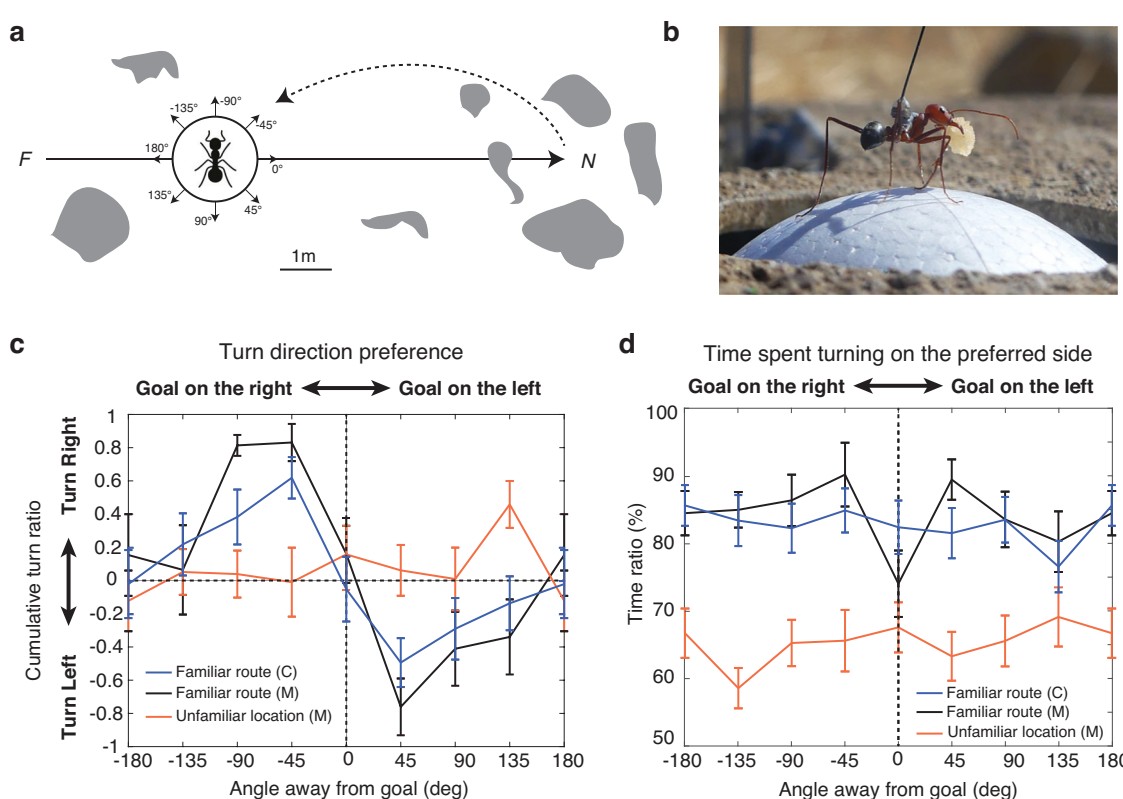

**Fig. 1 | Ants visually recognise whether the goal direction is left or right.**
**a** Homing ants were captured at the end of their familiar route and fixed on the trackball (**b**) in 8 different compass orientations. The route was rich in visual terrestrial cues (grey blobs). *F* feeder, *N* nest. **b** An individual *Cataglyphis velox* mounted on the trackball setup, holding its cookie crumb (credit Antoine Wystrach). **c** Turn ratio (*(right - left) / (right + left)*, with right/left angles derived from the integral of absolute angular velocity; mean ± se across individuals) for the eight compass directions, on the familiar route or in the unfamiliar location (same compass directions but unfamiliar surroundings) across 12 s of recording (see method). **d** Proportion of time spent turning on the preferred side of each individual (mean ± se across individuals). C *Cataglyphis velox* (n = 17), M *Myrmecia croslandi* (n = 11). Dynamics across time are provided in Supplementary Fig. 1. Statistics are provided in Supplementary Table. 1. Graphs in (**c**) and (**d**) were produced using MATLAB R2016b.

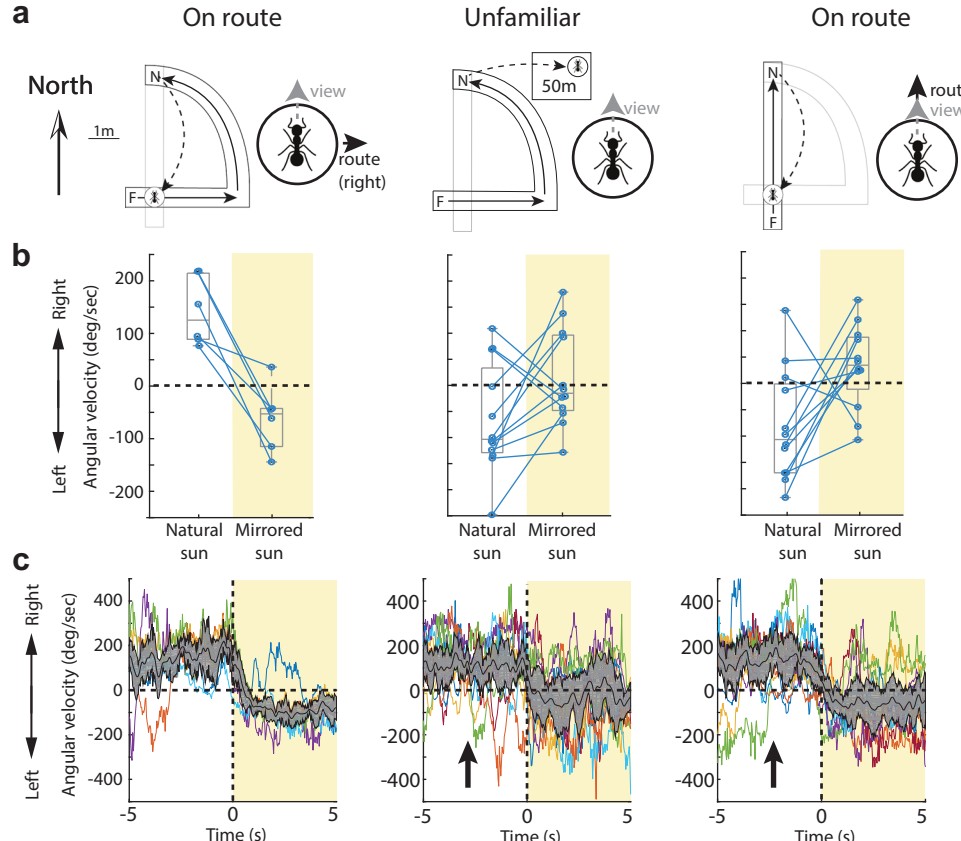

**Fig. 2 | Rotation of celestial cues shift turning direction based on familiar terrestrial cues. a** Schemes of the training and test condition. Homing ants were captured at the end of their familiar route (black arrows: familiar route, F feeder, N nest) and fixed on the trackball with their body always facing north, either on their route with the route direction 90° to the right (left panel); or within unfamiliar surroundings (middle panel); or ants were trained along a route oriented 90° to the previous one and released on their familiar route in the same location and orientation, which this time is facing their route direction. **b** Average angular velocity (positive = right turn) of each ant (dots) over the 5-s period before (white) and after (yellow) the apparent sun's position was mirrored by 180°. Wilcoxon test for 'turn towards the right with natural sun' (i.e., signed-rank test for angular velocity >0 with natural sun):

left panel: $n = 6$, $p = 0.0156$, power = 0.9981; middle panel: $n = 12$, $p = 0.9788$, power < 0.0001; right panel: $n = 12$, $p = 0.9866$, power < 0.0001. Wilcoxon test for 'turn direction reversal upon mirroring the sun' (i.e., signed-rank test for angular velocity <0 with paired 'natural sun × mirror') (left panel: $n = 6$, $p = 0.0156$, power = 0.9994; middle panel: $n = 12$, $p = 0.3955$, power = 0.4489; right panel: $n = 12$, $p = 0.0320$, power = 0.8127). Boxplots show median, 25th and 75th percentiles with whisker extending to extremum. **c** Turning velocities (individuals in colour; median ± iqr of the distribution in grey) across time, before and after the sun manipulation ($t_0$). Arrows in the middle and right panels: the velocities of some individuals have been inverted so that all individuals' mean turn directions before the manipulation are positive. Graphs in (**b**) and (**c**) were produced using MATLAB R2016b.

previously, makes a counterintuitive prediction: if the recognition of familiar views triggers a turn towards the correct goal-side (Fig. 1), reversing the direction of the compass representation in the CX should reverse the motor decision. We tested this prediction by mirroring the apparent position of the sun in the sky by 180° to ants tethered to our trackball system. The sun is an important source of celestial compass information, which feeds into the insects' CX[38,52,53], and previous studies have shown that this mirroring manipulation was sufficient to shift compass heading representation in desert ants[54,55], including the Cataglyphis species used here[13].

We first tethered well-trained ZV ants (i.e., captured just before entering the nest) on our trackball system with their body orientation fixed perpendicularly to their familiar route direction. As expected, ants in this situation turned towards the correct route direction (Fig. 2, left panels, natural sun), indicating that they correctly recognised familiar visual terrestrial cues. When mirroring the apparent sun's position by 180°, these ants responded by turning in the opposite direction within 1 s (Fig. 2, left panel, mirrored sun). We repeated the experiment by placing ZV ants in the same compass direction but in an unfamiliar location. In this situation, the ants turned in random directions (Fig. 2, middle panels), showing that the direction initially

chosen by the ants on their familiar route (Fig. 2, left panels) was based on the recognition of terrestrial rather than celestial cues. However, it remains unclear whether the sun's rotation had an impact on ants in unfamiliar terrain, as ants in this situation regularly alternate between left and right turns[26]. Finally, to ensure that the observed effect on route was not due to an innate bias at this particular location, we repeated this experiment with ants tethered at the exact same route location and body orientation, but trained to an alternative route, which was straight and aligned with the tethered direction of the trackball (Fig. 2, right panel). As expected, these ants facing their route heading showed no preference in turning direction at the group level, although most individuals still strongly favoured one side rather than walking straight (Fig. 2 right panels). Furthermore, mirroring the sun significantly reversed the individual's chosen direction, even though they were aligned with their goal direction (Fig. 2c right panels).

Taken together, these results demonstrate that guidance based on learnt views is a two-stage process: the recognition of visual memories—assumed to be through the MBs—does not directly drive motor commands, but it instead signals a desired heading—likely in the CX—, which in turn is used to control guidance using celestial compass information.

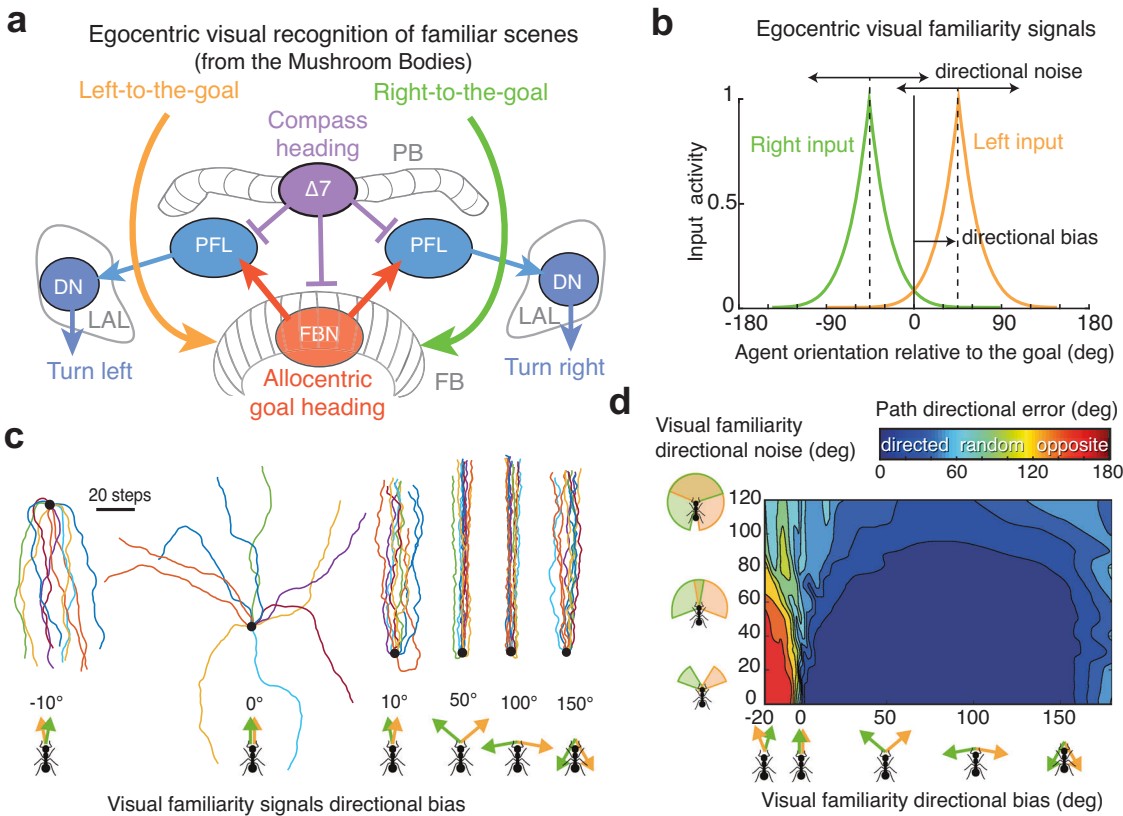

**Fig. 3 | Bilaterally decorrelated input to Central Complex produces stable route heading. a** The central complex (CX) sits in the brain midline but is wired to both hemispheres. It receives bilateral inputs in the Fan-Shaped-Body (FB), where sustained activity of the FB neurons (FBN) form an allocentric representations of the goal heading. PFL neurons−connecting the Protocerebral Bridge (PB) and FB to the Lateral Accessory Lobes (LALs)−compare this 'goal heading' with the current compass-based heading encoded by Δ7 neurons in the PB. They then produce lateralized outputs to the left and right LALs, modulating descending neurons (DNs) that control turning behaviour (see Supplementary Fig. 3). **b** Simulated inputs to the FBN neurons are body-orientation-dependant, mimicking visual familiarity[12] outputted by MBs[6]. 'Directional bias' indicates the angular offset from the goal direction (0°) at which the left visual familiarity signals are highest in average (e.g., +45° in this example). Right signal responds symmetrically (−directional bias). 'Directional noise' is modelled by randomly shifting the input response

curve around its mean (i.e., directional bias) at each time step, with values drawn from a normal distribution (see Supplementary Figs. 3 and 4). **c** Ten example simulated paths emerging from different directional biases (illustrated by green and orange arrows). **d** Resulting path directional error (absolute angular error between start-to-arrival and start-to-goal direction) after 200 steps, as a function of 'directional bias' (x axis) and 'directional noise' (y axis). **c, d** Stable route headings reliably emerge as long as left and right inputs signal when body-orientation is right and left from the goal, respectively (i.e., directional bias >0°) but not when both inputs peak when facing the goal (i.e., directional bias = 0°). Orientation towards the opposite direction emerges if left and right inputs signal inversely, that is, when the body is oriented right and left from the goal, respectively (i.e., directional bias <0°). Robustness to directional noise indicates that precise orientation during learning is not required. See further analysis in Supplementary Fig. 2. Graphs in (b−d) were produced using MATLAB R2016b.

## The CX circuitry is suited to receive lateralised input to produce goal-oriented paths

We next investigated how information about such a lateralised visual recognition of scenes−presumably computed in the MB−could plausibly be transmitted to the CX for guidance, given the known circuitry of insect brains. The CX's neural architecture is sufficiently mapped and conserved across species to allow biologically constrained neural modelling to address this question[39,56−58]. Research shows that the CX performs three key functions: (1) tracking the current heading in substructures such as the ellipsoid body (EB) and protocerebral bridge (PB)[35,38,59,60]; (2) retaining a goal heading representation for tens of seconds or more in the fan-shaped body (FB)[51,61,62]; and (3) comparing current and goal headings to produce compensatory left/right steering commands[39,51,60−64]. The goal heading can be updated via bilateral signals to the FB from external regions[51]. Such a signal may originate from the MB, directly or indirectly through the many relays observed in the dorsal Protocerebrum, notably the superior intermediate protocerebrum[5,29,50,51].

Our previous results indicate that the recognition of visual route memories−presumably in the MB−send lateralised information to a

centre processing compass information−presumably the CX−, signalling whether the goal is to the left or to the right. Interestingly, self-motion velocity signals sent to the CX, necessary for path integration, are also lateralised, with the left and right hemispheres transmitting velocities peaking at +45° and −45° offsets from the forward heading, respectively[39,65,66]. We thus built a model of the CX−as achieved previously for integrating velocity signals[39,57,65,66]−to test how this circuitry instead accommodates lateralised left and right signals from visual recognition in the MB (Fig. 3). The model was implemented in a simulated agent, creating a closed-loop system: the CX steers the agent heading exposing it to recognise left or right visual memories in the MB, which, in turn, updates the CX goal heading and influences subsequent steering decisions. Egocentric visual scene recognition in the MB is inherently noisy and sensitive to gaze orientation[6,12,67,68]. To account for this, we modelled visual recognition as requiring, at each time step, a precise alignment of the agent with a memorised 'offset' direction (i.e., the 'directional bias' in Fig. 3) to generate a strong familiarity signal. Departures from this gaze orientation lead to a rapid decay of the familiarity signal (Fig. 3b), as observed in Mushroom Body models[6,12,67,68]. As a result, views are often not recognised−or only

recognised with a low familiarity signal—resulting in sporadic, lateralised MB signals being transmitted to the CX during navigation.

The model produced stable route headings, showing the robustness of the CX in processing intermittent inputs to maintain navigational stability (Fig. 3). Impressively, varying parameters—such as the time during which FB neurons sustain their activity, motor or neural noise, the heading angle for which visual recognition is strongest, or large random variation around this angle—hardly had any effect on performance: straight routes emerged as long as left and right hemispheric inputs roughly correlated with a current right and left heading bias, respectively (Fig. 3, Supplementary Fig. 2). Conversely, reversing these correlations led to stable route-following in the opposite direction (Fig. 3). Having the MB signal correlating with moments where the agent faces the goal direction (directional bias 0° in Fig. 3c, d) corresponds to a labile zone of transition between two stable regimes of route-following in opposite directions. A thorough search through the parameter space revealed that this configuration produces a mediocre directionality at best, and is very sensitive to parameter change (Supplementary Fig. 2).

The robustness of the model with lateralised input was evident regardless of how the CX inner circuitry was modelled. Stable route-following emerged whether the CX was designed as in models of path integration in bees[39,57] (Supplementary Fig. 3) or allocentric travel tracking in flies[65,66] (Supplementary Fig. 4). This adaptability underscores the CX's capacity to form a stable goal heading when distinct lateralised input signals are present—one hemisphere signalling a rightward bias and the other a leftward bias. Note, however, that this does not imply that the MBs in each hemisphere exclusively compute left or right visual memories. Instead, both MBs likely process both left and right visual memories in different MB output neurons with opposing valences[69], the later being a characteristic feature of such neurons[70]. Lateralisation likely arises through opponent integration at downstream relay points[25,69,71], before being transmitted to the CX.

### A closed-loop interaction between egocentric terrestrial and allocentric celestial guidance

To further validate the proposed guidance system, we tested our model by simulating how agents navigating their familiar route would respond to a sudden 135° shift in the CX's compass estimate of the current heading (Fig. 4). We then compared these simulated behaviours to those of real homing ZV *C. velox* ants in an analogous scenario, where the sun's position was artificially shifted by 135° using a mirror as the ants recapitulated their familiar route.

The results highlight a complex closed-loop interaction between the use of long-term terrestrial memory cues and heading estimates derived from compass information. Notably, the sudden shift in the sun's position triggered the ants to veer immediately, confirming that—even while on a familiar route—their instantaneous guidance is at least partly influenced by their internal compass representation. After walking a few centimetres in the wrong direction, the ants reoriented to realign with the correct route, indicating that terrestrial cues provided corrective directional information (Fig. 4a, green).

Interestingly, the corrective turn was followed by a period of meandering, suggesting transient uncertainty in directional control (Fig. 4b, green). Remarkably, despite the nonlinear dynamics at play, the simulated responses closely matched those observed in the ants under sun manipulation (Fig. 4a, b, blue). The period of directional uncertainty in the model emerges from a superimposition of different goal directions in the FB of the CX: the previous goal direction (now misaligned with the route given the updated compass representation) overlaps with the newly updated goal direction based on the novel compass frame. The agent resumes a straight path as the old goal memory trace in the FB decays (Fig. 4c). The fact that ants resume a straight path within seconds suggests a relatively fast decay of this memory trace.

Overall, these results bolster the model's credibility and offer valuable insights into the mechanisms underlying insect navigation.

## Discussion

Insects and vertebrates alike use a combination of egocentric (viewpoint-dependent) and allocentric (world-referenced) spatial representations for visual navigation. However, how both are combined in the vertebrate brain is far from clear[5,50,51,72–74]. Here, we showed how egocentric recognition of visual scenes can be converted into an allocentric goal heading in the insect brain, and how both interact as a closed-loop to produce robust navigation. The advantages of such a transfer of reference frame are at least twofold. First, egocentric visual recognition enables a remarkably compact storage of long-term visual memories but is highly sensitive to gaze orientation[6,67], whereas an allocentric representation enables steering in the direction of travel independently of body orientation[39,65,66]. The transfer from the former to the later offers a higher order flexibility, and explains, for instance how ants can visually recognise their route forward—using their MB[3–5]—and subsequently follow such a direction backwards—using the CX[13]. Second, recognition of natural scenes in the MB is inherently noisy and sensitive to visual obstructions, which must happen continuously when navigating through grassy or leafy environments[13,75]. In contrast, the CX provides a stable and sustained allocentric heading representation by integrating self-motion[35] with multiple wide-field celestial[38] and terrestrial cues[37,76]. The CX can thus act as a buffer between noisy sensory recognition and smooth motor control. Overall, this architecture combines the strengths of an opponent system—where comparing left and right memories signal reduces sensitivity to absolute fluctuation in visual familiarity—with those of an allocentric buffer system, which maintains smooth and holonomic motor control despite sporadic visual recognition. A neural model of the MB and CX implementing this design produces remarkably robust navigation in realistic environments[69]. This design shows how insects' flexible navigational skills can arise from a rigid, egocentric encoding of visual memories in the MB. How rigid this visual encoding truly is, and whether the MB can, to some extent, extract spatial structure or support partial rotational invariance, remains an open question.

The lateralised design shown here, where visual memories indicate whether the goal is on the left- or right-hand side rather than straight ahead, is present in the two phylogenetically and ecology distant ant species we tested, which suggests an interesting take on the evolution of navigation in bilateral animals. Segregating 'turn left' and 'turn right' signals between hemispheres recalls tropotaxis, where orientation is achieved by comparing signal intensities from left and right sensors[77–83]. The conserved arthropods' CX may well be evolutionarily constrained to process such lateralised inputs, as observed for self-motion velocity signals[39,65,66]. The evolution of visual route-following in hymenopterans is relatively recent and cannot function with direct comparisons of left and right visual sensors. Probably as a consequence, each eye projects to both hemispheres' MBs[84,85] making visual scene memories mainly binocular[86]. Categorising learnt binocular views when facing 'left' or 'right' from the goal, and segregating this information across hemispheres may thus reflect an evolutionary adaptation to align visual processing with the CX's bilateral input requirements.

How left and right visual memories are acquired remains to be seen. Our model shows that the gaze angle at which views are learnt does not need to be precisely controlled (Fig. 3c, d). This fits with the observation that naive insects exploring the world for the first time expose their gaze in all directions—and not only in the nest or anti-nest direction—providing ample opportunities to form a rich set of left-to-the-goal and right-to-the-goal visual memories[16,19,24,69,87], therefore enabling corrective turns from most positions. The first source of information about whether the current body orientation is left or right from the goal probably results from path integration. Lateralised

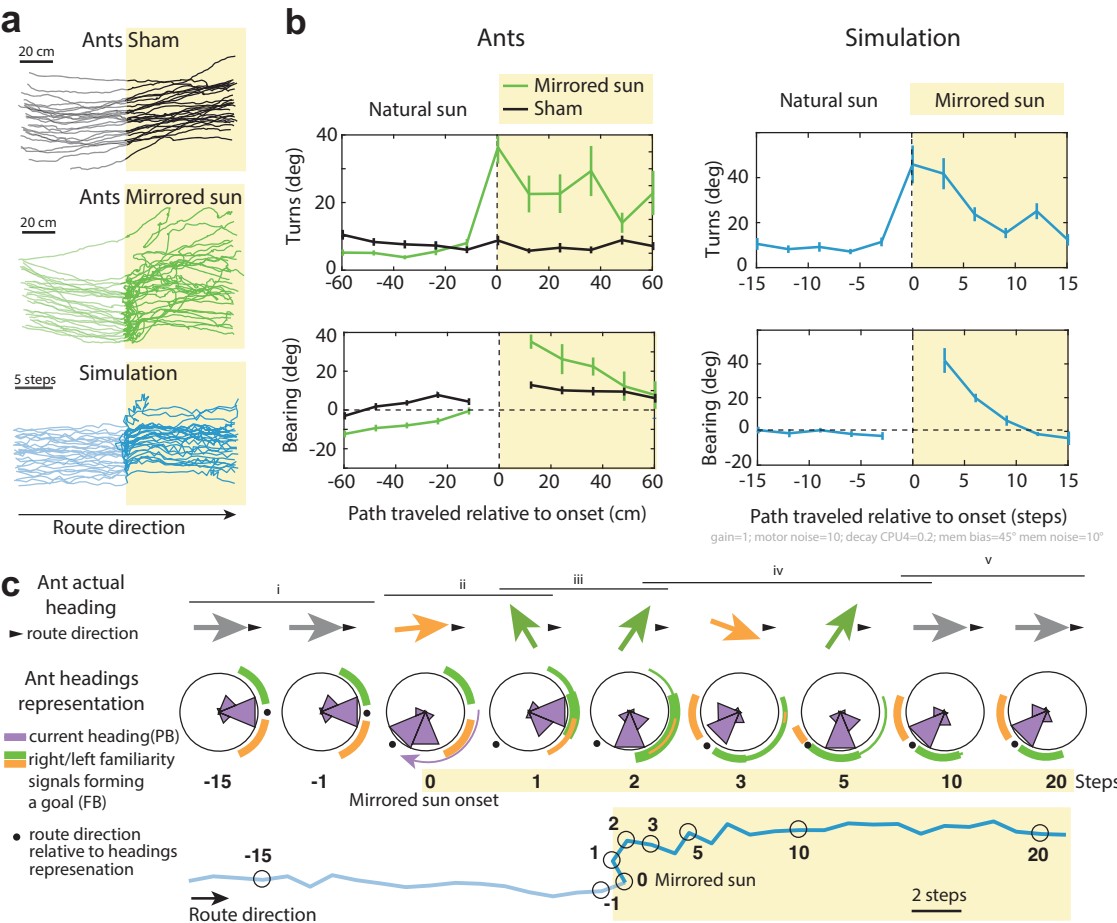

**Fig. 4 | Rotation of celestial cues disrupts route following as predicted by the model. a** Paths of real ants (black and green) and simulated ants (blue) retracing a familiar straight route as they enter a zone where celestial compass cues were manipulated (yellow). All ants were in a zero-vector state (i.e., path integration was non-informative). In the 'Mirrored Sun' condition (green, $n = 29$), the real sun was hidden and reflected to appear 135° counterclockwise in the sky. In the 'Sham' condition (black, $n = 26$), the sun was also blocked, but only a small region of sky—excluding the sun—was mirrored. Simulated ants ($n = 20$) followed the model from Fig. 3, where sun rotation was implemented as a 135° shift in the current heading signal (a 3-cell bump shift in the Protocerebral Bridge). **b** Quantification of bearings (relative to the route direction, 0°) and turn angles (absolute angle between successive segments), showing means ± SE before and after sun rotation onset. Real ant paths were discretized every 12 cm; simulated paths every 3 steps. Turns at $x = 0$ correspond to changes at the moment of celestial cue rotation. **c** The model's behaviour depends on parameters (e.g., gain = 1, motor noise = 10°/step, FBN decay = 0.2, visual familiarity bias ± noise = 45° ± 10°; see Supplementary Fig. 2), but its qualitative dynamics remain robust: (i) Initially, heading is stabilised between left and right goal representations in the Fan-shaped Body (FB), updated by lateralised familiarity signals. (ii) Sun rotation causes a shift in the current heading representation (PB), leading to a mismatch with the stored FB goal direction and triggering a left turn. (iii) This deviation exposes the ants to a 'left-view', which activates the opposite side's FB, inducing a right-turn. (iv) The incoming familiarity signals create a conflicts between new and old goal direction, causing meandering. (v) Eventually, the new goal direction stabilises as memory traces of the old familiarity signals fade. See Supplementary Fig. 3 for detailed dynamics. Graphs in (**a**–**c**) were produced using MATLAB R2016b.

dopaminergic feedback from the lateral accessory lobes (LAL)—a premotor area receiving path integration outputs—to specific compartments of the MBs could represent an ideal candidate to orchestrate such a categorisation of left/right memories[69,88]. Indeed, the insect brain receives massive lateralised signal from motor feedback[89,90], including LAL to MB neurons (see for instance[91]). Revisiting current questions in insect and robot navigation, such as early exploration, route following and homing[17,18,31,92,93]; the integration of aversive memories[25,94,95], lateral oscillations[46,96–98], or multi-modal integration[99–106] with such a lateralised design as a framework promises an interesting research agenda.

## Methods

### The trackball setup

For experiments shown in Figs. 1 and 2, we used the air-suspended trackball setup as described in Dahmen et al., 2017[44]; and chose the configuration where the ants are fixed in a given direction and cannot physically rotate (if the ant tries to turn, the ball counter-rotates under

its legs). To fix ants on the ball, we used a micro-magnet and metallic paint applied directly on the ant's thorax. The trackball air pump, battery and computer were connected to the trackball through 10 m long cables and hidden in a remote part of the panorama. The trackball movements were recorded using custom software in C++, data was analysed with Matlab R2016b.

### Route setups and ant training in Cataglyphis velox

For all experiments (Figs. 1, 2 and 4), *C. velox* ants were constrained to forage within a route using half-sunken wood planks that prevented their escape, while leaving the surrounding panoramic view of the scenery intact (as described in Wystrach et al., 2012[107]). Nest to feeder distances were 10 m (2 m wide corridor) in experiment 1 (Fig. 1), 4 m (1 m wide corridor for the straight part) in experiment 2 (Fig. 2), and 10 m (2 m wide corridor) in experiment 3 (Fig. 4). Cookie crumbs were provided ad libitum in the feeder positions for at least 2 days before any tests. Some barriers dug into the ground created baffles, enabling us to control whether ants were experienced with the route. Ants were

considered trained when able to home along the route without bumping into any such obstacle. These ants were captured just before they entered their nest to ensure that they could not rely on path integration (so-called ZV ants), marked with a metallic paint on the thorax and a colour code for individual identification, and subjected to tests (see next sections).

## Route setups and ant training in *Myrmecia croslandi*

For the experiment with *M. croslandi* ants (Fig. 1), we used each individual's natural route, for which these long-lived ants have extensive experience[41]. Individuals were captured on their foraging trees, marked with both metallic paint and a colour code for individual identification, given a sucrose solution or a prey and released where they had been captured (on their foraging tree). Upon release, these ants immediately started to return home. We followed them while marking their route using flag pins every 50 cm (so that their exact route was known). We captured the ants just before they entered their nests and subjected them to the test, with the trackball (see next section) placed exactly on their route (thanks to the flags) at a distance of at least 8 m away from their nest. Total route lengths varied from 10 to 27 m and were overall straight.

Experimental protocol for the left/right trackball experiment (Fig. 1):

- An experienced ant was captured just before entering its nest and marked with a drop of metallic paint on the thorax.
- A large opaque ring (30 cm diameter, 30 cm high) was set around the trackball setup.
- The ant was fixed on the trackball within the opaque ring, which prevented her to see the surroundings. Only a portion of the sky above was accessible to the ant.
- The trackball system (together with the opaque ring and the fixed ant within) was moved to the desired position and rotated so that the ant was facing the desired direction.
- One experimenter started recording the trackball movements (from the remote computer), when another lifted the ring (so the ant could see the scenery) before leaving the scene, letting the ant behave for at least 15 s post ring lifting.
- The experimenter came back, replaced the ring around the trackball system, and rotated the trackball system (following a pre-established pseudo-random sequence) for the ant to face in a novel direction.
- We repeated steps 5 and 6 until the 8 possible orientations were achieved (the sequence of orientations was chosen in a pseudo-random order so as to counter-balance orientation and direction of rotation).

The data shown in Fig. 1 for each orientation is averaged across 12 s of recording (from 3 to 15 s post-ring lifting). We decided to let 3 s after the ring lifting, because the experimenter's movements before leaving the area might disturb the ants.

In all experiments, Cataglyphis ants were tested only once. In Experiment 1, Myrmecia ants were tested successively in familiar and unfamiliar terrain, in randomised order.

Experimental protocol for the mirror trackball experiments (Fig. 2):

- An experienced ant was captured just before entering its nest and marked with a drop of metallic paint on the thorax.
- A large opaque ring (30 cm diameter, 30 cm high) was set around the trackball setup.
- The ant was fixed on the trackball within the opaque ring, which prevented her to see the surroundings. Only a portion of the sky above was accessible to the ant.
- The trackball system (together with the opaque ring and the fixed ant within) was moved to the desired position and rotated so that the ant was facing the desired direction.

- One experimenter started recording the trackball movements, when another lifted the ring (so the ant could see the scenery) before leaving the scene, letting the ant behave for at least 10 s post ring lifting.
- Two experimenters simultaneously hid the real sun using a 50 × 50 cm wood board, and projected the reflected sun using a mirror, so that the sun appeared in the opposite half of the sky relative to the ant (as in ref. 13), for at least 8 s.

Ants were tested only once, in one of the conditions. For the two-legged route, individual attribution to the conditions 'On route' and 'Unfamiliar' were interleaved. The straight route condition was achieved subsequently as it require to train a different cohort of individuals. Note that the present and following experiment could not be achieved with Myrmecia ants, whose compass system, probably adapted to the forest environments where the sun is often occluded, is rather insensitive to the sun's position. The compass system of the desert ant *C. velox*, on the other hand, relies heavily on the sun's position[13].

## Experimental design and protocol for the mirror experiment with ants on the floor (Fig. 4)

*C. velox* ants were trained to a 10 m-long route (route corridor 2 m wide) for at least two consecutive days. A 240 × 120 cm thin wood board was placed on the floor in the middle of the route, ensuring that the navigating ants walked smoothly without encountering small clutter over this portion of the route. Homing ants were captured just before entering their nest and released at the feeder as ZV ants. Upon release, these ZV ants typically resume their route homing behaviour; at mid-parkour (halfway along the board section), the real sun was hidden by one experimenter and reflected by another, using a mirror, for the sun to appear to the ant 135° away from its original position in the sky. To ensure that each individual was tested only once, tested ants were marked with a drop of paint after the procedure.

The ZV ants walking on the board were recorded using a Panasonic Lumix DMC-FZ200 camera on a tripod, and their paths were digitised frame by frame at 10 fps using image J. We used four marks on the board to correct for the distortion due to the tilted perspective of the camera's visual field. Analysis of the paths was achieved with Matlab R2016b 2016.

## The central complex neural models

The CX models circuitries and input signals are described in Supplementary Figs. 3 and 4, and the different parameters used to obtain the output (motor command) are described in Supplementary Fig. 2. All the modelling has been achieved with Matlab R2016b 2016.

## Statistics and reproducibility

No statistical method was used to predetermine sample size. We expected strong, dichotomous effects that would clearly verify our a priori predictions (left vs. right turns on the trackball; Figs. 1 and 2; left turns accompanied by a sharp increase in sinuosity upon mirroring the sun; Fig. 4). Therefore, we used a relatively small sample size and non-parametric statistical tests, aiming to detect robust effects while minimising the likelihood of false positives.

No individuals were excluded from the analyses. Experimental conditions were randomised as described in the respective protocols (see 'Experimental protocol for the left/right trackball experiment,' Fig. 1; and 'Experimental protocol for the mirror trackball experiments,' Fig. 2). Data consisted of video tracking (Fig. 4) and sensor recordings of trackball movements (Figs. 1 and 2), which were automatically captured and thus not subject to investigator bias during recording.

## Reporting on sex

Because ant foragers are only females, our study is only using female subjects.

**Reporting summary**

Further information on research design is available in the Nature Portfolio Reporting Summary linked to this article.

## Data availability

The data generated in this study have been deposited on GitHub: https://github.com/antnavteam/lateralised_visual_memories.

## Code availability

Codes to analyse data and run simulations have been deposited on GitHub: https://github.com/antnavteam/lateralised_visual_memories.

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

## Acknowledgements

We thank Profs. Jochen Zeil and Xim Cerda to provide us access to field sites in ANU Canberra, Australia and Seville, Spain, respectively. We are grateful to Prof. Hansjuergen Dahmen for helping us setting the trackball device. We thank the Profs. Rüdiger Wehner, Tom Collett and Paul Graham for fruitful discussions and comments on earlier versions of the manuscript. This work was funded by the ERC Starting Grant EMERG-ANT no. 759817 and ERC Consolidator Grant RESILI-ANT no. 101125881 to AW.

## Author contributions

Research design: A.W. Data collection: A.W., S.S., F.L.M. and L.C. Trackball system development: F.L.M. Data analysis: A.W., S.S. and F.L.M. Modelling: A.W. Manuscript writing: A.W.

## Competing interests

The authors declare no competing interests.
