## [Transparent Peer Review file · Nature Communications]

Updating an allocentric goal from lateralised egocentric visual memories

Corresponding Author: Dr Antoine Wystrach

Version 0:

Reviewer comments:

Reviewer #1

(Remarks to the Author)

In this work the authors use a clever combination of controlled behavioural experiments in the ants' natural environment and computational modelling. They demonstrate that ants, suspended on a trackball to prevent their ability to explore in translation or rotation, show a lateralized steering response towards a familiar route when placed at a range of orientations on the route. Next they show that rotating the allocentric cue (i.e. the sun), leads to a corresponding rotation in the initial trajectory of these ants. Hence, the allocentric celestial compass is involved in egocentric guidance based on memorised visual cues. The authors go on to demonstrate how noisy, lateral inputs corresponding to visual familiarity and associated with a turning command, together with global cues, allow for a central complex circuitry based model resulting in directed control. This is robust to a range of parameters and also accurately reflects the shift experienced in the trajectory of real ants exposed to a rotation in the celestial cue. This is exciting and significant research, however, the modeling requires more explanation and it is not clear how a fundamental change in behaviour (directed backward facing routes) is achieved between the main work and the supplementary materials (see comments below). Overall, this work puts forward a novel, biologically plausible and parsimonious strategy for visual route following. We only have sparse knowledge how brains achieve the remarkable navigational behaviour observed in insects and this work significantly improves our understanding how the brains of insects integrate information from egocentric and allocentric cues for navigation. Please see the specific comments below.

Main text:

L33-35: This assumes that ants are unable to perform mental rotation, as being able to do so would provide an indication as to whether to turn left or right. Ardin et al (Frontiers 2016) don't rule out mental rotation when considering the ability of ants to home when travelling backward. Either this needs to be considered as a possibility for generating lateral commands, or its discounting should be appropriately justified.

L34: Requires unpacking of what recognition means and what the steps of this process are, explaining visual recognition signals (as later appearing in L60).

L50-54 & L86-88: It seems that this response at 0 degrees could be because the route is not perfectly straight. Especially if distinct left/ right responses are triggered for individual ants but are not apparent on a global scale. The routes of ants are tortuous and idiosyncratic (e.g. Kohler & Wehner 2005, Mangan & Webb 2012). Are these ants placed at orientations towards the 'average' overall direction of the route, but might they, on a smaller scale, actually be oriented with a rotational offset to locally learned views? Particularly given the statements in lines L191-193 (and also to some degree L14-15 in the abstract) on naive ants. These ants exhibit exploratory routes and, as stated, 'expose their gaze in all directions', perhaps as a result of this exploratory route structure.

L52: Should probably be figure 1d?

L107: Explain what 'SIP' is.

L119: how did you model visual recognition, what was the structure of the proxy like?

L118-121: How are 'sporadic signals transmitted to the CX', if there is a strong familiarity signal at 'each time step'? This

sentence requires clarification. Also, it is also not entirely clear what 'fortunate alignment' means.

L108-121: Additionally to the above, the modelling could use more explanation. For example, figure 3b needs to be briefly explained in the text. Additionally, some of the explanations in the supplementary materials are better suited to the main body to aid in explanation of the modelling, for example, some of the parameter explanation (L33-63 in supplementary materials) and some of the information in extended figure 2 could be briefly highlighted here.

L147-157: Very little comment on the results, needs more description. What are the nonlinear dynamics at play? Why is it remarkable? How is the model in agreement with the experimental results?

L191-193: For clarity here it would be useful to reiterate that left/ right visual memories are tied to signaling which way to turn, rather than a 'step forward' (e.g., as in L176-177). Perhaps when leading into line L194, as this feels like a jump.

L216: There is more information about the data analyses required in the method sections. For example: (1) For figure 1: How did you calculate (right-left)/(right+left)? Did you define full turns and then calculated if they were left or right turns or did you add up the angular changes between frames? Also, say how you defined 'preferred side'. (2) For figure 2: Over what time period did you calculate angular velocity in b and c? (3) For figure 4b: Were turns and bearings calculated over a segment? You have to briefly say how you have calculated these.

L228-237: Please provide the nest-to-feeder distance for each experiment. How wide were the routes?

L240: Should probably be figure 1 instead of 2.

L242: How far away were these trees?

L245-246: State more clearly why you had to know their exact route. Did you track the routes as well or did you simply do some measurements while the flags were there?

L249: Please say in this section how far away from the nest you placed the trackball.

L267: Should be figure 1.

L270: Ants were tested either at unfamiliar or familiar sites but tested in all directions. Hence, ants were tested multiple times. I would thus slightly rephrase this to make it clearer.

L284: What did you use to hide the sun?

L284-286: I suggest citing here a reference where this was done before.

L287: See comment for L270.

Figures

Figure 1

a/b: A panoramic image representing the ants' view would aid this figure.

c/d: Examples of the path from -90, -45, 0, 45 etc. would assist this figure, however, this could also go in the supplementary information. Also, would there be a way to add some statistics here that backs up your conclusion?

Figure 2

L523: Clarification on 5s is required, is this instantaneous or over a period?

L524-527: The wording 'turn towards the right...' is confusing (e.g. in the middle or right panel they don't have a preference to turn right, isn't?). Did you test if data is different from 0? It needs to be much clearer what exactly you have tested and about which panel you are talking about. As this is paired data, it seems like an opportunity for a pairwise comparison between each of the pairs of 'natural' and 'mirrored' sun. Only one statement of 'power' in the stats, rather than throughout stats in 2b.

c: Arrows are not in the panels described (middle and left as opposed to middle and right), not specifically pointing at the reversed trajectory velocities.

Figure 3

a: DNs needs to be explained, was the intention MNs?

c: need to say that the yellow/ orange arrows represent the directional bias (if this is the case!).

Between b/c: To aid in a more step by step explanation, a subfigure perhaps like that of the top right panel of extended 2.

Figure 4

b: you need to say if this is the mean. What do the error bars represent? Also, would there be a way to add some statistics here that backs up your conclusion?

c: Black dot in ant heading representation is confusing, perhaps a different marker and a title in the key saying 'route direction according to celestial compass'?

c: Colours behind arrows are confusing, as the yellow is very similar to the background colour when the sun has been rotated. Remove or modify.

c: You need to explain the width and height of the yellow and orange bars around the ant heading representation diagrams

(circles).

L570: for consistency, counter clockwise or left?

L573: wrong figure reference (Figure 3, not 1)

L577-578: more details required on segment analysis, what does a segment consist of?

Supplementary Materials

Extended figure 1:

The colour bar covers the same colour ranges as in figure 3, but now indicates 'directed' (blue) to 'random' (red). For consistency with figure 3, use the range 'directed' (blue) to 'random' (green).

Importantly, what is the difference between extended figure 1 and figure 3 from the main paper? What is the difference in implementation or parameter which means that the opposite paths don't occur anymore? This is not clear, the explanation in L10-18 requires more detail.

Explain the difference between 'visual recognition directional basis (average)' in extended figure 1a and 'visual recognition directional basis' in main figure 3d.

Extended figure 2:

Top right panel labelling is confusing, use subheadings to denote the steps and keep a, b, c etc for main panels of this figure.

See also the comments for figure 4 in main text.

Be a little bit more specific how the yellow and orange histograms relate to the arced bars in the circular representation in (e).

L72: as observed in other insects, please specify.

L73: 'spaced' 45 degrees apart, or similar phrasing, for clarification.

L99-100: 'Numbers on the left indicate neuron numbers', please clarify (e.g. 'where brain areas are preceded by a number, this indicates the number of neurons')

Extended figure 3:

3f is not in agreement with 3g. If 3g is correct, why doesn't the backward behaviour happen here but in the paths shown in 3f?

As above, top right panel labelling is confusing, use subheadings to denote the steps.

Why is 'average' missing from the x axis title in 3g?

L139: HΔB, PFNs etc. are not defined, make sure any new regions or abbreviations are introduced.

General

Lines 4, 10, 12 13, 44, 79, 131. Referring to figure 1 in main text, when this should be figure 3.

Some typos for main and extended:

L40: ensure → ensured

L42: cues → cue

L47: was → were

L49: orientation → orientations

L68: feed → feeds

L96: produces → produce

L113: signal → signals

L113/114: accommodates instead → instead accommodates

L119: as → at

L159: brackets instead of these hyphens

L166: enables to steer the direction ... → enables steering in the direction...

L184: eyes → eye

L198: feedbacks → feedback

L229: them to → their

L267/268: sec → seconds

Figure 3, L546: bracket

Figure 4, L583: situation → situations

Supplementary:

L151: ...neurons (DN), which difference... → ...neurons (DN), for which difference...

We hope these comments will help improve your paper. Nice work!

Amany Amin & Cornelia Buehlmann

Reviewer #2

(Remarks to the Author)

I co-reviewed this manuscript with one of the reviewers who provided the listed reports. This is part of the Nature Communications initiative to facilitate training in peer review and to provide appropriate recognition for Early Career

Researchers who co-review manuscripts.

Reviewer #3

(Remarks to the Author)

The manuscript by Wystrach et al is a very interesting combination of behavioral experiments and computational modeling that provides surprising novel insight into the possible neural mechanisms underlying ant navigation. It addresses how view based memories are integrated with vectorial representations of traveling directions to enable robust route following. The experiments are elegant and combine track-ball setups used outdoors in the natural environment of desert ants to investigate how these insects respond to familiar and unfamiliar scenes presented at specific angles relative to their body axis. The main result is that ants appear to recognize familiar views irrespective of their own orientation, but respond with distinct steering movements when the familiar scene is towards the right or left compared to their goal heading along the familiar route. This elegantly combines view recognition and celestial compass navigation, gives a first idea of why two memory centers (mushroom bodies) are needed in the brain, but only one central complex (i.e. an explanation for lateralization), and provides a clear concept of how insects can use the same steering circuits for highly different navigational strategies.

The predictions from the presented computational models are nicely checked against real data and make this model highly convincing. The emerging concepts are also relevant beyond ants and illuminate how neural systems can evolve new navigation strategies despite constraints imposed by ancient circuit components (or in fact by taking advantage of them). The presented circuit level hypotheses are testable by experiments at the neural level and thus have the potential to move the field forwards significantly. Overall, the work is of high quality, clear writing and nicely illustrated. I have two main points below that need to be addressed, as well as several minor points, but generally think that this paper would be a suitable contribution for Nature Communications.

Major comments

1. The experiments are directly interpreted in a neural context and the text leaves the impression that the neural conclusions directly follow from the data. This is however not the case, as no neural manipulations are presented that allow the confirmation of the involvement of the CX and the MB in the processes described. While I agree that the explanations are perfectly in line with the current understanding of these brain regions, and likely are not incorrect, these points are still only predictions that match the data, and no direct conclusions from the data. The authors should thus change the language of the text to adjust accordingly. E.g. it states: "...results indicate that MB recognition of route memories send lateralized information to the CX." The results do not show that information is sent to the CX. While likely, other possibilities also exist in the brain and cannot be ruled out without direct experimental manipulations of the involved neural pathways. This would of course be far beyond the scope of the paper, and the issue can be fixed with careful rewording. Similar statements can be found throughout the paper.

2. It is currently not fully clear what the second ant species adds to the paper, especially as only experiment 1 was carried out with both, and the other figures only show *Cataglyphis* data. It should be clarified why the second species was used for experiment 1 and why this was not needed for the remaining experiments. Showing that the identified mechanisms are generally important across distantly related ants is important in my view, but this idea should be highlighted more if that point should come across as a main conclusion.

Minor comments

1. Minor grammar edits needed throughout the paper.

2. Extended data figure 3, schematic network diagram: EPG neurons are only described as excitatory in the literature, but here are both excitatory and inhibitory. Maybe replace with delta7 (also indicating heading), as those have been shown to be mostly inhibitory, but also excitatory towards one PFN neuron type in flies (so that would fit the model very nicely).

3. Extended data figure 2: Using fly neuron names in one figure and the old bee neuron names in another is confusing. I recommend using the fly names consistently, as these will be more widely known in the field.

4. Modelling figures: Yellow lines and fonts are nearly impossible to read on a print out version of the paper.

5. line 205: remove 'the' before Prof Hansjürgen Dahmen

Reviewer #4

(Remarks to the Author)

Decades of remarkable field studies in different species of foraging ants have revealed their use of path integration and route following in navigating away from and then back to their nests. Meanwhile, anatomical and physiological insight from studies, mainly in other insects, have highlighted the how specific highly conserved insect brain regions, particularly the central complex (CX) and mushroom body (MB), might play complementary roles in implementing such algorithms. Models inspired by behavioral, physiological, and anatomical studies have suggested specific computations for circuits within these

brain regions, but these models have yet to be fully tested. Specifically, the CX is known to carry out head-direction or compass-based computations while the MB provides the substrate for associative memory that could include some form of visual-template learning. This manuscript uses a fascinating and clever open-loop track-ball setup in the field to record the behavioral responses of ants to views at familiar locations along their foraging route. The authors show that views trigger directional turning responses which seek to align the animal to the route direction. Further, the directionality of the responses is dependent on the location of an allocentric (global) sun-compass cue in the ant's visual field. This, along with known anatomy and prior experimental work, enabled the authors to build a computational model that utilizes a feedforward signal from the MB to the CX. The experimental data and the modeling work in this manuscript have the potential to be of considerable interest to researchers working in animal behavior and studying the neuroscience of navigation, although there are some issues that deserve further attention before publication.

The authors argue that the left and right MBs (L and R MBs) store views that the ant experiences to the left and the right of the learnt route direction at a given location ("left view" and "right view"). To determine which direction to face at a given location along the ant's return journey, the current view is independently compared with the left and right views which generate two independent familiarity signals that are then passed on to the central complex (CX), specifically the fan shaped body (FB) neurons (also known as PFNs). These independent left and right familiarity signals can then be used by extensively modeled CX circuitry to compute a turning direction. Overall, the behavioral data is novel in its approach and findings and certainly indicates the presence of a feedforward signal from the MB to the CX. However, the data seem insufficient to suggest that the MB stores left and right views, which are independently compared with the current view presented to the ant. The manuscript might benefit by modifying the model to explicitly account for these gaps.

Major Concerns

In the experiment, the ant is "clamped" to one of 8 directions at a single familiar location along its route and its turning response is recorded for a 20 second duration. The authors find that the presentation of a view not along the route direction directs turns that seek to align the animal with the route. The claim that the MBs store left and right views at every point along the route is not uniquely (or strongly enough) supported by the behavioral data. It is possible that the ant stores a single view in the direction of the route across synapses in both the L and R MB, and that the comparison of the current view with this single stored view contains directional information. This might still lead to distinct familiarity signals in the L and R MBs which could drive the L and R PFNs differently as proposed in the model. The manuscript might benefit from explicit descriptions, if possible, of the scenes viewed by the ant at the given experimental location to clarify whether directional information can be extracted by a comparison of a side view with the view in the route direction. Further, explicitly modeling how the MB signal is computed from the scene (as opposed to proposing a form for the signal relative to the route direction) might provide useful insight into the possible ways in which views are stored and compared in the MB, and exactly how these different ways either account for the observed data or fall short.

Minor Concerns

Figure 1: "Cumulative turn ratio" instead of "Cumulated turn ratio"?

Methods don't contain a description of the turn ratio. Why was 12 seconds chosen for averaging?

How is goal direction experimentally determined? Is it the direction towards the nest? Is it the direction tangential to the route? Are these the same for the chosen route? From previous work one would imagine it's tangential to the route, but the methods section should be more explicit about this as well.

Here it might be useful to see the shape of the route.

33: Lent 2010, 2013 suggest ants can compute turn direction based on the presented view. This should be cited and discussed.

48: The sinusoidal curve in Fig. 1c can be explained without the need for lateralized signals from the MB to the PFNs. The existence of some goal direction vector in the FB aligned with the route direction should produce the curve shown. It would be useful to plot the turn ratio for each individual ant in the unfamiliar location to see if it has a similar curve which is not centered at the route direction. In general, individual data points might be more informative than averaged curves.

It might also be useful to analyze behavior conditioned on the order in which the ant was exposed to each of the 8 directions. Does the turn size (mean angular velocity) change as the ant samples each orientation? Is the change in turn size predicted based on which view the ant was previously exposed to?

49: orientations instead of orientation

65: Cartwright and Collett (1983) suggest that landmark navigation might depend on the allocentric reference frame. This should be cited.

82: remove extra "ants"

141: Doesn't it still requires two separate plasticity events at the same location when the ant is facing left and when it is facing right of the route? How would this happen?

184-185: Unclear if individual KCs pool over multiple VPNs and what the proportion of ipsilateral dominant vs contralateral dominant KCs is.

198: Motor feedback to the MB from the CX in general might not be sufficient, it should be specific to the DANs. Is there evidence in the connectivity for this?

Reviewer #5

(Remarks to the Author)

Version 1:

Reviewer comments:

Reviewer #1

(Remarks to the Author)

Thank you for addressing the comments in our previous review, we are certainly satisfied with the changes and clarifications provided and raise only a few minor comments:

L45-48: These lines are a helpful addition but they need a little bit of clarification/elaboration in my view.

Is the sentence beginning 'however', and in particular 'short-term mapping', referring to path integration as calculated with terrestrial cues? I think it could help to relate this to what I think you mean by 'path integration' in the next sentence (i.e path integration in general).

With respect to 'different surroundings', perhaps clarifying that you mean somewhere distinct from their collection point but still familiar e.g 'distinct yet familiar surroundings'?

Typos/ minor clarifications:

L44: 'without other source' -> without any other source/ without other sources/ without sources other than the terrestrial view (for more clarification).

L596: The part 'Turn ratio (integral of absolute angular velocity: $(\text{right} - \text{left}) / (\text{right} + \text{left})$...)' looks a little like the formula relates to the integral rather than the turn ratio, perhaps clarification with e.g. ' $(\text{right} - \text{left}) / (\text{right} + \text{left})$, with right/left angles derived from the integral of absolute angular velocity'

L647: This should probably say 'green and orange arrows'.

L680, 689 & 691: check that colours in legend and figure are the same.

L681: just a slight rewording to 'yellow and orange marks, for which extent and thickness illustrate respectively the agent's recently sampled directions and familiarity signals received'

Supplementary materials:

Figure 1 is certainly a helpful addition, one minor point would be to consider the issue with the formula in line 7 (as discussed for the main text above).

L113: should say c)

Reviewer #3

(Remarks to the Author)

In the revised version of the manuscript the authors have addressed all my previous concerns. I have only three more minor points.

1. As far as I understand, a coordinating role of the LAL in locking turning signals to specific phases of an ant's motor oscillations has also been proposed in a recent review paper (Collett, T., Graham, P. and Heinze, S. (2025). The neuroethology of ant navigation. *Current Biology* 35, R110–R124.), maybe worth including in line 241 in the discussion.

2. The dual role of delta7 cells, i.e. an ability to excite some neurons and inhibit other downstream neurons, was suggested via physiological methods in Franconville, R., Beron, C. and Jayaraman, V. (2018). Building a functional connectome of the *Drosophila* central complex. *eLife* 7, e04577. This paper should be cited to justify the modeling choice shown in extended data figure 4.

3. The headings on top of the pages of the extended figures are out of sync with the figure legends (figure 3 is referred to as

figure 2, and figure 4 is referred to as figure 3)

Reviewer #4

(Remarks to the Author)

The authors have made changes that clarify some of the issues that we raised. These largely address the main concerns, but some additional edits would also be helpful. Specific points below:

In their first answer, the authors suggest that the views experienced by the ant are stored in a “fundamentally egocentric way” — i.e. that a static view cannot be used by the ants to extract an appropriate turning direction. To convince us they cite 3, 19, 20, 21, all of which propose that the parsimonious explanation for ant behavior is a view-matching strategy in which the ant only has access to a single familiarity value in each direction at one point along the route. They suggest that this is why they need stored views in multiple directions at each point along the route and the reason why ants need to scan left or right to perform route navigation. This is consistent with the result that ants carrying their food backwards cannot recognize a scene along a route they usually take in the forward direction. They also point out that the ants experience multiple views at each point along the route providing lots of opportunities to store “left of goal” and “right of goal” views.

These are all fair points. However, the authors' argument still relies on parsimony and does not rule out the suggested alternative. In other words, IF the ant can compare two distinct familiarity values from the left and right MB for a single gaze direction to arrive at an instantaneous signed turning signal, this would also be consistent with the experimental results cited. This is why we requested that the authors show some of the scenes viewed by the ants at a few locations. If that is not feasible, we would suggest that the authors include a single sentence, perhaps in General Discussion if not Results, stating that further experimentation is required to definitively rule out the possibility that the comparison of an instantaneous view with a single stored snapshot at each point along the route is used by insects to immediately estimate which direction to turn.

For Answer 2, the citation in the text needs to be to the authors' bioRxiv manuscript, not Aso et al., 2014 (which is what reference 69 seems to point to).

For Answer 8, the authors agree that the sinusoidal curve in Fig 1c can be explained without the need for lateralized signals from the MB to the PFNs. They should update the text to include that possibility.

For Answer 9, Extended data fig 1 only shows data for a familiar location, but it would be useful to also see it for an unfamiliar location, as we suggested.

With the few changes above, the manuscript would be appropriate for publication without further review.

Reviewer #5

(Remarks to the Author)

Dear editor,

We are very grateful to the four reviewers for their time, enthusiasm, and constructive feedback. We have incorporated their suggestions into the revised manuscript, added an additional figure (Extended Data Fig. 1), and included new statistical analyses. All changes in the text are highlighted in yellow. Below, you will find our detailed responses (in bold) to each comment.

With best wishes,

Antoine

REVIEWER COMMENTS

Reviewer #1 (Remarks to the Author):

In this work the authors use a clever combination of controlled behavioural experiments in the ants' natural environment and computational modelling. They demonstrate that ants, suspended on a trackball to prevent their ability to explore in translation or rotation, show a lateralized steering response towards a familiar route when placed at a range of orientations on the route. Next they show that rotating the allocentric cue (i.e. the sun), leads to a corresponding rotation in the initial trajectory of these ants. Hence, the allocentric celestial compass is involved in egocentric guidance based on memorised visual cues. The authors go on to demonstrate how noisy, lateral inputs corresponding to visual familiarity and associated with a turning command, together with global cues, allow for a central complex circuitry based model resulting in directed control. This is robust to a range of parameters and also accurately reflects the shift experienced in the trajectory of real ants exposed to a rotation in the celestial cue. This is exciting and significant research, however, the modeling requires more explanation and it is not clear how a fundamental change in behaviour (directed backward facing routes) is achieved between the main work and the supplementary materials (see comments below). Overall, this work puts forward a novel, biologically plausible and parsimonious strategy for visual route following. We only have sparse knowledge how brains achieve the remarkable navigational behaviour observed in insects and this work significantly improves our understanding how the brains of insects integrate information from egocentric and allocentric cues for navigation. Please see the specific comments below.

Answer: Thanks for your enthusiasm! You will find detailed answers to your concerns (including the backward facing routes) below.

Main text:

L33-35: This assumes that ants are unable to perform mental rotation, as being able to do so would provide an indication as to whether to turn left or right. Ardin et al (Frontiers 2016) don't rule out mental rotation when considering the ability of ants to home when travelling backward. Either this needs to be considered as a possibility for generating lateral commands, or its discounting should be appropriately justified.

+ L34: Requires unpacking of what recognition means and what the steps of this process are, explaining visual recognition signals (as later appearing in L60).

Answer: Indeed. We added a small section in the first paragraph of the introduction to clarify what we meant by 'visual recognition', and simultaneously discount mental rotation. *"These visual memories are stored in an egocentric way, requiring the insects –despite their close to full-panoramic*

visual field¹⁶⁻¹⁸ –to align their gaze direction as experienced previously to recognise a view^{3,19-21}. For instance, ants can no longer recognise a one-way familiar route if their gaze is oriented away from the usual direction, such as when dragging a heavy food item backwards²². This calls into question their ability to perform mental rotations of the perceived scene, and explains why ants, wasps and bees need to physically ‘scan’ multiple directions to recognise memorised views and guide their path accordingly²²⁻²⁶. The observation that these insects regularly ‘turn back and look’ while departing from their nest or a feeder led to the dominant idea they memorise views while facing the goal^{22,26-32} (or anti-goal³³⁻³⁵), and must consequently align their body in these same directions to recognise a learnt view as familiar when subsequently returning to the goal^{21,29,34,36-42}. This is parsimonious as view familiarity indicates that the goal is ahead and can simply trigger a ‘go forward’ command^{39,43}. However, it implies that recognition bears no indication as to whether to turn left or right^{21,29,34,36-42}. “

L50-54 & L86-88: It seems that this response at 0 degrees could be because the route is not perfectly straight. Especially if distinct left/ right responses are triggered for individual ants but are not apparent on a global scale. The routes of ants are tortuous and idiosyncratic (e.g. Kohler & Wehner 2005, Mangan & Webb 2012). Are these ants placed at orientations towards the ‘average’ overall direction of the route, but might they, on a smaller scale, actually be oriented with a rotational offset to locally learned views? Particularly given the statements in lines L191-193 (and also to some degree L14-15 in the abstract) on naive ants. These ants exhibit exploratory routes and, as stated, ‘expose their gaze in all directions’, perhaps as a result of this exploratory route structure.

Answer: Agreed. We added the following sentence: “Perhaps it is possible to identify, for each individual, a precise orientation—slightly offset from the general route and nest direction—that triggers purely forward movement. However, we note that even when ants are free to recapitulate their familiar route on the ground, they continuously alternate between left and right turns rather than proceed straight³⁷⁻⁴¹.”

L52: Should probably be figure 1d?

Answer: Yes, thank you

L107: Explain what ‘SIP’ is.

Answer: done. “Such signal may originate from the MB, directly or indirectly through the many relays observed in the dorsal Protocerebrum, notably the Superior Intermediate Protocerebrum (SIP)^{1-3,38}.”

L119: how did you model visual recognition, what was the structure of the proxy like?

+ L118-121: How are ‘sporadic signals transmitted to the CX’, if there is a strong familiarity signal at ‘each time step’? This sentence requires clarification. Also, it is also not entirely clear what ‘fortunate alignment’ means.

+L108-121: Additionally to the above, the modelling could use more explanation. For example, figure 3b needs to be briefly explained in the text. Additionally, some of the explanations in the supplementary materials are better suited to the main body to aid in explanation of the modelling,

for example, some of the parameter explanation (L33-63 in supplementary materials) and some of the information in extended figure 2 could be briefly highlighted here.

Answer: Indeed, we opted for a brief description of the model and left the technical details out of the main rationale. This is because those details are directly drawn from previous literature and are not essential to understanding the core, novel contribution of our study. Given the broad readership of *Nature Communications*, we felt it was better to keep the main text concise and focused (as you noted, the technical details are provided elsewhere). Nonetheless, in light of your comment, we have now added clarifications to the main text regarding how visual recognition is modelled: “we modelled visual recognition as requiring, at each time step, a precise alignment of the agent with a memorised ‘offset’ direction (i.e., the ‘directional bias’ in Fig. 3) to generate a strong familiarity signal. Departures from this gaze orientation lead to a rapid decay of the familiarity signal (Fig. 3b), as observed in Mushroom Body models^{7,21,67,68}. As a result, views are often not recognised—or only recognised with a low familiarity signal—resulting in sporadic, lateralised MB signals being transmitted to the CX during navigation.”

L147-157: Very little comment on the results, needs more description. What are the nonlinear dynamics at play? Why is it remarkable? How is the model in agreement with the experimental results?

Answer: The detailed explanation was included in figure 4’s legend to save space. I now unpacked the description directly in the main text:

“The results highlight a complex closed-loop interaction between the use of long-term terrestrial memory cues and heading estimates derived from compass information. Notably, the sudden shift in the sun’s position triggered the ants to veer immediately, confirming that—even while on a familiar route—their instantaneous guidance is at least partly influenced by their internal compass representation. After walking a few centimetres in the wrong direction, the ants reoriented to realign with the correct route, indicating that terrestrial cues provided corrective directional information (Fig. 4a, green).

Interestingly, this was followed by a period of meandering, suggesting transient uncertainty in directional control (Fig. 4b, green). Remarkably, despite the nonlinear dynamics at play, the simulated responses closely matched those observed in the ants under sun manipulation (Fig. 4a, b, blue). The period of directional uncertainty in the model emerges from a superimposition in the FB of the CX: the previous goal direction (now misaligned with the route given the updated compass representation) overlaps with the newly updated goal direction based on the novel compass frame. The agent resumes a straight path as the old memory trace in the FB decays (Fig. 4c). The fact that ants resume a straight path within seconds suggests a relatively fast decay of this memory trace.

Overall, these results bolster the model’s credibility and offer valuable insights into the mechanisms underlying insect navigation.”

L191-193: For clarity here it would be useful to reiterate that left/ right visual memories are tied to signaling which way to turn, rather than a ‘step forward’ (e.g., as in L176-177). Perhaps when leading into line L194, as this feels like a jump.

Answer. Done “providing ample opportunities to form a rich set of left-to-the-goal and right-to-the-goal visual memories^{25,28,33,70,84}, therefore enabling corrective turns from most positions.”

L216: There is more information about the data analyses required in the method sections. For example: (1) For figure 1: How did you calculate (right-left)/(right+left)? Did you define full turns and then calculated if they were left or right turns or did you add up the angular changes between frames? Also, say how you defined 'preferred side'. (2) For figure 2: Over what time period did you calculate angular velocity in b and c? (3) For figure 4b: Were turns and bearings calculated over a segment? You have to briefly say how you have calculated these.

Answer. This info is now indicated directly in the figure legends (it has been added in fig. 1, and was already present for fig. 2 and 3). All are highlighted in yellow in the new manuscript version.

L228-237: Please provide the nest-to-feeder distance for each experiment. How wide were the routes?

Answer: Done. "Nest to feeder distances were 10 m (2 m wide corridor) in experiment 1 (Fig. 1), 4 m (1 m wide corridor for the straight part) in experiment 2 (Fig.2), and 10 m (2 m wide corridor) in experiment 3 (Fig. 4)."

L240: Should probably be figure 1 instead of 2.

Answer: Indeed, thanks.

L242: How far away were these trees?

+L245-246: State more clearly why you had to know their exact route. Did you track the routes as well or did you simply do some measurements while the flags were there?

+ L249: Please say in this section how far away from the nest you placed the trackball.

Answer: Information added: "We captured the ants just before they entered their nests and subjected them to the test, with the trackball (see next section) placed exactly on their route (thanks to the flags) at a distance of at least 8 metres away from their nest. Total route lengths varied from 10 to 27 metres and were overall straight."

L267: Should be figure 1.

Answer: Indeed, thanks.

L270: Ants were tested either at unfamiliar or familiar sites but tested in all directions. Hence, ants were tested multiple times. I would thus slightly rephrase this to make it clearer.

Answer. Done. "In Experiment 1, *Myrmecia* ants were tested successively in familiar and unfamiliar terrain, in randomised order."

L284: What did you use to hide the sun?

Answer: Added. “Two experimenters simultaneously hid the real sun using a 50 x 50 cm wood board, and projected the reflected sun using a mirror, so that the sun appeared in the opposite half of the sky relative to the ant (as in ²²), for at least 8 seconds.”

L284-286: I suggest citing here a reference where this was done before.

Done

L287: See comment for L270.

Answer: Left unchanged as ants were tested only once.

Figures

Figure 1

a/b: A panoramic image representing the ants' view would aid this figure.

c/d: Examples of the path from -90, -45, 0, 45 etc. would assist this figure, however, this could also go in the supplementary information. Also, would there be a way to add some statistics here that backs up your conclusion?

Answer: We have now added an extended data figure (4) that includes the ants' turning dynamics across time. These are more informative than paths (which are hard to interpret as ants were fixed on the trackball, and the paths typically loop over themselves). We also added a supplementary table of the desired statistics. We unfortunately do not have panoramic pictures of the ants' viewpoints.

Figure 2

L523: Clarification on 5s is required, is this instantaneous or over a period?

Done: “Average angular velocity (positive = right turn) of each ant (dots) over the 5-second period before (white) and after (yellow) the apparent sun's position was mirrored by 180°.”

L524-527: The wording ‘turn towards the right...’ is confusing (e.g. in the middle or right panel they don't have a preference to turn right, isn't?). Did you test if data is different from 0? It needs to be much clearer what exactly you have tested and about which panel you are talking about. As this is paired data, it seems like an opportunity for a pairwise comparison between each of the pairs of ‘natural’ and ‘mirrored’ sun.

Answer: We tested for whether the ants ‘turned towards the right with natural sun’, with a Wilcoxon signed rank test, which amount to test whether the average angular velocity is > 0, which is now mentioned “(signed-rank test for angular velocity < 0)”. This first test is not for paired data, as it concerns here only the ‘natural sun’ condition (as indicated). This is just to ensure that the first group is oriented towards the right, and that the two controls (middle and right panel), are not. Note: the next test is necessarily individually paired, as we mention testing for ‘turn direction reversal’.

Only one statement of ‘power’ in the stats, rather than throughout stats in 2b.

Answer: Done for the 3 groups x 2 tests in the legend of Figure 2.

c: Arrows are not in the panels described (middle and left as opposed to middle and right), not specifically pointing at the reversed trajectory velocities.

Answer: "middle and right", thanks.

Figure 3

a: DNs needs to be explained, was the intention MNs?

Answer: indeed, corrected, thanks

c: need to say that the yellow/ orange arrows represent the directional bias (if this is the case!).

Answer: Indeed, done

Between b/c: To aid in a more step by step explanation, a subfigure perhaps like that of the top right panel of extended 2.

Answer: We prefer stowing this in the extended data to keep the main manuscript length manageable. Also, showing both inputs across 360° on a same graph provides a better representation of how the azimuths are covered. In case doubts persist for some readers, I added in the figure legend panel b: "(see top right panel of Extended Data fig. 3, 4 for a broken-down representation)".

Figure 4

b: you need to say if this is the mean. What do the error bars represent?

Added "(mean ± se)".

Also, would there be a way to add some statistics here that backs up your conclusion?

Answer: Designing a statistical test in this case would be somewhat post-hoc, as the experiment was not originally designed to test a specific behavioural hypothesis, but rather to observe the emerging dynamics and assess whether a model could qualitatively capture some of them. It is also worth noting that the difference between the Sham and Mirrored Sun conditions is visually apparent in both the trajectories and the quantified data shown here (as evidenced by the very large separation between standard error bars in Fig. 4b).

c: Black dot in ant heading representation is confusing, perhaps a different marker and a title in the key saying 'route direction according to celestial compass'?

c: Colours behind arrows are confusing, as the yellow is very similar to the background colour when the sun has been rotated. Remove or modify.

c: You need to explain the width and height of the yellow and orange bars around the ant heading representation diagrams (circles). L573: wrong figure reference (Figure 3, not 1)

L577-578: more details required on segment analysis, what does a segment consist of?

Answer: All done, thanks

L570: for consistency, counter clockwise or left?

Answer: Clockwise (or counterclockwise) when it refers to a rotation, left or right when it refers to a direction.

Supplementary Materials

Extended figure 1:

The colour bar covers the same colour ranges as in figure 3, but now indicates 'directed' (blue) to 'random' (red). For consistency with figure 3, use the range 'directed' (blue) to 'random' (green).

Answer: Reducing the colour range from blue (0°) to green (90°) would strongly decrease the readability of panels b and c, of which ranges are small. Also, it is important to understand that 90° is the maximum possible value here (contrary to figure 3), so the full colour range should reflect this. See next comment for more explanation.

Importantly, what is the difference between extended figure 1 and figure 3 from the main paper? What is the difference in implementation or parameter which means that the opposite paths don't occur anymore? This is not clear, the explanation in L10-18 requires more detail.

Answer: We now clarified the explanation: "a. Same as Fig. 3d, except that, here, the search explores the entire parameter space to find regimes minimizing path directional error. In Fig. 3d, unrepresented parameters (motor noise, motor gain, memory decay) were fixed at average values, which led to routes in the opposite direction (i.e., directional error ~180°) when directional bias < 0. In contrast, the full parameter searches here can select combinations (typically with high motor noise) that produce very tortuous paths—resulting in random directions (directional error ~90°) – rather than routes in opposite directions (~180°). This shows that when visual familiarity bias is < 0, no parameter regime can yield straight, goal-oriented paths (directional error ~0°)"

Explain the difference between 'visual recognition directional basis (average)' in extended figure 1a and 'visual recognition directional basis' in main figure 3d.

Answer: They are the same. "(average)" has now been removed to avoid confusion

Extended figure 2:

Top right panel labelling is confusing, use subheadings to denote the steps and keep a, b, c etc for main panels of this figure.

See also the comments for figure 4 in main text.

Be a little bit more specific how the yellow and orange histograms relate to the arced bars in the circular representation in (e).

Answer: All done, thanks

L72: as observed in other insects, please specify.

L73: 'spaced' 45 degrees apart, or similar phrasing, for clarification.

L99-100: 'Numbers on the left indicate neuron numbers', please clarify (e.g. 'where brain areas are preceded by a number, this indicates the number of neurons')

Answer: All done (added references for the 'other insects')

Extended figure 3:

3f is not in agreement with 3g. If 3g is correct, why doesn't the backward behaviour happen here but in the paths shown in 3f?

Answer: This was because the paths were achieved with fixed parameters, and the map was achieved by exploring throughout the parameter space. The map has been remade using fixed parameters (as in fig. 3), making both consistent. Thanks for spotting this.

As above, top right panel labelling is confusing, use subheadings to denote the steps.

Answer: Done

Why is 'average' missing from the x axis title in 3g?

Answer: Average has been removed everywhere

L139: HΔB, PFNs etc. are not defined, make sure any new regions or abbreviations are introduced.

Answer: Done

General

Lines 4, 10, 12 13, 44, 79, 131. Referring to figure 1 in main text, when this should be figure 3.

Answer: Done, thanks!

Some typos for main and extended:

L40: ensure → ensured

L42: cues → cue

L47: was → were

L49: orientation → orientations

L68: feed → feeds

L96: produces → produce

L113: signal → signals

L113/114: accommodates instead → instead accommodates

L119: as → at

L159: brackets instead of these hyphens

L166: enables to steer the direction ... → enables steering in the direction...

L184: eyes → eye

L198: feedbacks → feedback

L229: them to → their

L267/268: sec → seconds

Figure 3, L546: bracket

Figure 4, L583: situation → situations

Answer: All fixed, thanks

Supplementary:

L151: ...neurons (DN), which difference... → ...neurons (DN), for which difference...

Answer: Done

We hope these comments will help improve your paper. Nice work!

Amany Amin & Cornelia Buehlmann

Answer: Your comments definitely helped improving the manuscript, thanks heaps for your time and accurate read.

Thank you both very much for your detail read and helpful comments.

Reviewer #2 (Remarks to the Author):

Reviewer #3 (Remarks to the Author):

The manuscript by Wystrach et al is a very interesting combination of behavioral experiments and computational modeling that provides surprising novel insight into the possible neural mechanisms underlying ant navigation. It addresses how view based memories are integrated with vectorial representations of traveling directions to enable robust route following. The experiments are elegant and combine track-ball setups used outdoors in the natural environment of desert ants to investigate how these insects respond to familiar and unfamiliar scenes presented at specific angles relative to their body axis. The main result is that ants appear to recognize familiar views irrespective of their own orientation, but respond with distinct steering movements when the familiar scene is towards the right or left compared to their goal heading along the familiar route. This elegantly combines view recognition and celestial compass navigation, gives a first idea of why two memory centers (mushroom bodies) are needed in the brain, but only one central complex (i.e. an explanation for lateralization), and provides a clear concept of how insects can use the same steering circuits for highly different navigational strategies.

The predictions from the presented computational models are nicely checked against real data and make this model highly convincing. The emerging concepts are also relevant beyond ants and illuminate how neural systems can evolve new navigation strategies despite constraints imposed by ancient circuit components (or in fact by taking advantage of them). The presented circuit level hypotheses are testable by experiments at the neural level and thus have the potential to move the field forwards significantly. Overall, the work is of high quality, clear writing and nicely illustrated. I have two main points below that need to be addressed, as well as several minor points, but generally think that this paper would be a suitable contribution for Nature Communications.

Major comments

1. The experiments are directly interpreted in a neural context and the text leaves the impression that the neural conclusions directly follow from the data. This is however not the case, as no neural manipulations are presented that allow the confirmation of the involvement of the CX and the MB in the processes described. While I agree that the explanations are perfectly in line with the current understanding of these brain regions, and likely are not incorrect, these points are still only predictions that match the data, and no direct conclusions from the data. The authors should thus change the language of the text to adjust accordingly. E.g. it states: "...results indicate that MB recognition of route memories send lateralized information to the CX." The results do not show that information is sent to the CX. While likely, other possibilities also exist in the brain and cannot be

ruled out without direct experimental manipulations of the involved neural pathways. This would of course be far beyond the scope of the paper, and the issue can be fixed with careful rewording. Similar statements can be found throughout the paper.

Answer: You are absolutely right. We now corrected these parts, which now read for instance:
“Our previous results indicate that the recognition of visual route memories – presumably in the MB – send lateralised information to a centre processing compass information – presumably the CX –”

or

“Taken together, these results demonstrate that guidance based on learnt views is a two-stage process: the recognition of visual memories – assumed to be through the MBs – does not directly drive motor commands, but it instead signals a desired heading – likely in the CX “.

2. It is currently not fully clear what the second ant species adds to the paper, especially as only experiment 1 was carried out with both, and the other figures only show *Cataglyphis* data. It should be clarified why the second species was used for experiment 1 and why this was not needed for the remaining experiments. Showing that the identified mechanisms are generally important across distantly related ants is important in my view, but this idea should be highlighted more if that point should come across as a main conclusion.

Answer: Indeed, we now mention the reason in the method: *“Note that the present and following experiment could not be achieved with *Myrmecia* ants, whose compass system, probably adapted to the forest environments where the sun is often occluded, is rather insensitive to the sun’s position. The compass system of the desert ant *Cataglyphis velox*, on the other hand, relies heavily on the sun’s position²²”*

Also, we have leveraged the use of two species in the discussion to give more credence to the evolutionary scenario proposed. *“The lateralised design shown here, where visual memories indicate whether the goal is on the left- or right-hand side rather than straight ahead, is present in the two phylogenetically and ecology distant ant species we tested, which suggests an interesting take on the evolution of navigation in bilateral animals.”*

Minor comments

1. Minor grammar edits needed throughout the paper.

Answer: Indeed, we have now performed a careful re-read.

2. Extended data figure 3, schematic network diagram: EPG neurons are only described as excitatory in the literature, but here are both excitatory and inhibitory. Maybe replace with delta7 (also indicating heading), as those have been shown to be mostly inhibitory, but also excitatory towards one PFN neuron type in flies (so that would fit the model very nicely).

Answer: Excellent point, thank you.

3. Extended data figure 2: Using fly neuron names in one figure and the old bee neuron names in another is confusing. I recommend using the fly names consistently, as these will be more widely known in the field.

Answer: Done

4. Modelling figures: Yellow lines and fonts are nearly impossible to read on a print out version of the paper.

Answer: Yellow has been replaced by a darker green

5. line 205: remove 'the' before Prof Hansjürgen Dahmen

Answer: Done

Thank you very much for your appreciation and helpful comments.

Reviewer #4 (Remarks to the Author):

Decades of remarkable field studies in different species of foraging ants have revealed their use of path integration and route following in navigating away from and then back to their nests. Meanwhile, anatomical and physiological insight from studies, mainly in other insects, have highlighted the how specific highly conserved insect brain regions, particularly the central complex (CX) and mushroom body (MB), might play complementary roles in implementing such algorithms. Models inspired by behavioral, physiological, and anatomical studies have suggested specific computations for circuits within these brain regions, but these models have yet to be fully tested. Specifically, the CX is known to carry out head-direction or compass-based computations while the MB provides the substrate for associative memory that could include some form of visual-template learning. This manuscript uses a fascinating and clever open-loop track-ball setup in the field to record the behavioral responses of ants to views at familiar locations along their foraging route. The authors show that views trigger directional turning responses which seek to align the animal to the route direction. Further, the directionality of the responses is dependent on the location of an allocentric (global) sun-compass cue in the ant's visual field. This, along with known anatomy and prior experimental work, enabled the authors to build a computational model that utilizes a feedforward signal from the MB to the CX. The experimental data and the modeling work in this manuscript have the potential to be of considerable interest to researchers working in animal behavior and studying the neuroscience of navigation, although there are some issues that deserve further attention before publication.

The authors argue that the left and right MBs (L and R MBs) store views that the ant experiences to the left and the right of the learnt route direction at a given location ("left view" and "right view"). To determine which direction to face at a given location along the ant's return journey, the current view is independently compared with the left and right views which generate two independent familiarity signals that are then passed on to the central complex (CX), specifically the fan shaped body (FB) neurons (also known as PFNs). These independent left and right familiarity signals can then be used by extensively modeled CX circuitry to compute a turning direction. Overall, the behavioral data is novel in its approach and findings and certainly indicates the presence of a feedforward signal from the MB to the CX. However, the data seem insufficient to suggest that the MB stores left and right views, which are independently compared with the current view presented to the ant. The manuscript might benefit by modifying the model to explicitly account for these gaps.

Major Concerns

In the experiment, the ant is "clamped" to one of 8 directions at a single familiar location along its route and its turning response is recorded for a 20 second duration. The authors find that the

presentation of a view not along the route direction directs turns that seek to align the animal with the route. The claim that the MBs store left and right views at every point along the route is not uniquely (or strongly enough) supported by the behavioral data. It is possible that the ant stores a single view in the direction of the route across synapses in both the L and R MB, and that the comparison of the current view with this single stored view contains directional information. This might still lead to distinct familiarity signals in the L and R MBs which could drive the L and R PFNs differently as proposed in the model. The manuscript might benefit from explicit descriptions, if possible, of the scenes viewed by the ant at the given experimental location to clarify whether directional information can be extracted by a comparison of a side view with the view in the route direction.

Answer: Yes, this requires some clarifications. As you mentioned, our main conclusion is agnostic to how left and right signal arise from the MB. Nonetheless, we modelled familiarity as assuming multiple stored memories, which –contrary to the assumption that the ant could derive a direction from a current view misaligned with its memory – is in phase with the previous behavioural and neurological literature. We now emphasize in the introduction the previous literature showing that ants store and recognise views in a fundamentally egocentric way, and cannot derive directional information when misaligned with their stored views.

“These visual memories are stored in an egocentric way, requiring the insects –despite their nearly full-panoramic visual field^{16–18} –to align their gaze direction with what they experienced previously in order to recognise a view^{3,19–21}. For instance, ants can no longer recognise the natural scene along a familiar one-way route if their gaze is oriented away from the usual direction, such as when dragging a heavy food item backwards²². This calls into question their ability to perform mental rotations of the perceived scene and explains why ants, wasps, and bees need to physically ‘scan’ multiple directions to recognise memorised views and guide their path accordingly^{22–26}.”

Also, we give more details onto why we modelled visual recognition as multiple memory banks:

“Egocentric visual scene recognition in the MB is inherently noisy and sensitive to gaze orientation^{7,21,67,68}. To account for this, we modelled visual recognition as requiring, at each time step, a precise alignment of the agent with a memorised ‘offset’ direction (i.e., the ‘directional bias’ in Fig. 3) to generate a strong familiarity signal. Departures from this gaze orientation lead to a rapid decay of the familiarity signal (Fig. 3b), as observed in Mushroom Body models^{7,21,67,68}. As a result, views are often not recognised—or only recognised with a low familiarity signal—resulting in sporadic, lateralised MB signals being transmitted to the CX during navigation.”

Finally, we emphasize in the discussion some aspects of behaviour which further back up the idea that ants do not only learn views that aligned with or away from the nest direction:

“This fits with the observation that naive insects exploring the world for the first time expose their gaze in all directions –and not only in the nest or anti-nest direction– providing ample opportunities to form a rich set of left-to-the-goal and right-to-the-goal visual memories^{25,28,33,70,84}, therefore enabling correct corrective turns from most positions.”

Further, explicitly modeling how the MB signal is computed from the scene (as opposed to proposing a form for the signal relative to the route direction) might provide useful insight into the possible ways in which views are stored and compared in the MB, and exactly how these different ways either account for the observed data or fall short.

Answer: We fully agree. However, given the scope and length of implementing a full-fledged MB neural model—and importantly by the fact that it is not necessary to support our main conclusions in the first place—we have treated this task separately. Specifically, we developed a neural MB and CX model embedded in an agent operating within a 3D environment. This constitutes a separate paper, currently available on bioRxiv, which we now reference more explicitly in the Discussion section: “A neural model of the MB and CX implementing this design produces remarkably robust navigation in realistic environments⁶⁹”

Minor Concerns

Figure 1: “Cumulative turn ratio” instead of “Cumulated turn ratio”?

Answer: Indeed, done

Methods don't contain a description of the turn ratio.

Answer: It is now explained in the figure legend. Turn ratio (*integral of absolute angular velocity: (right - left) / (right + left)*);

Why was 12 seconds chosen for averaging?

Answer: This is explained in the method “The data shown in Figure 1 for each orientation is averaged across 12 seconds of recording (from 3 to 15 seconds post- ring lifting). We decided to let 3 seconds after ring lifting, because the experimenter’s movements before leaving the area might disturb the ants.”

How is goal direction experimentally determined? Is it the direction towards the nest? Is it the direction tangential to the route? Are these the same for the chosen route? From previous work one would imagine it's tangential to the route, but the methods section should be more explicit about this as well. Here it might be useful to see the shape of the route.

Answer: Figure 1a shows the relation between the trackball and the route. We also added a description of the route for both species used, in the method. “Nest to feeder distances were 10 m (2 m wide corridor) in experiment 1 (Fig. 1), 4 m (1 m wide corridor for the straight part) in experiment 2 (Fig.2), and 10 m (2 m wide corridor) in experiment 3 (Fig. 4).”

33: Lent 2010, 2013 suggest ants can compute turn direction based on the presented view. This should be cited and discussed.

Answer:

Indeed, we now cite both references. In the introduction: “It should be noted that ants and flies subjected to a sudden jump of the surrounding scene without other source of compass information can correct their course accordingly^{12,51}.”

And later: “Perhaps it is possible to identify, for each individual, a precise orientation that triggers purely forward movement. However, we note that even when ants are free to recapitulate their familiar route on the ground, they continuously alternate between left and right turns rather than proceed straight⁴⁷⁻⁵¹”

48: The sinusoidal curve in Fig. 1c can be explained without the need for lateralized signals from the MB to the PFNs. The existence of some goal direction vector in the FB aligned with the route direction should produce the curve shown.

Answer: Absolutely, and the subsequent mirror experiment (fig. 2) is in agreement with this. The current experiment (fig. 1) shows that visual scene recognition can update/set this goal direction vector in the FB.

It would be useful to plot the turn ratio for each individual ant in the unfamiliar location to see if it has a similar curve which is not centered at the route direction. In general, individual data points might be more informative than averaged curves.

Answer: Agreed. We now show individual data in Extended data fig. 1.

It might also be useful to analyze behavior conditioned on the order in which the ant was exposed to each of the 8 directions. Does the turn size (mean angular velocity) change as the ant samples each orientation? Is the change in turn size predicted based on which view the ant was previously exposed to?

Answer: We agree that this would be interesting to investigate, however it would require another experimental design. Indeed, here the orientation sequence was pseudo-randomised so that no two ants underwent the same sequence among the 4030 (factor 8) possible ones (times 2 if we take into account the actual direction of rotation, which was also randomised). Also, each orientation*sequence_number was equally represented across the tested population. As a result of this design we have therefore very little data per possible transition, or orientation*sequence number, which makes it impossible to conduct meaningful statistics about such interactions; thus preventing to draw conclusions about the effects of the previously exposed orientation.

141: Doesn't it still requires two separate plasticity events at the same location when the ant is facing left and when it is facing right of the route? How would this happen?

Answer: We assume this is achieved in different MB output neurons, so separate plasticity events in different location. We clarified the sentence: "Instead, both MBs likely process both left and right visual memories in different MB output neurons with opposing valences, a characteristic feature of such neurons⁶⁹."

184-185: Unclear if individual KCs pool over multiple VPNS and what the proportion of ipsilateral dominant vs contralateral dominant KCs is.

Answer: indeed, this quantification has not yet been achieved in these ant species.

198: Motor feedback to the MB from the CX in general might not be sufficient, it should be specific to the DANs. Is there evidence in the connectivity for this?

Answer: Indeed, such dopaminergic feedback from the LAL targeting specific compartment of the mushroom body exists. We now emphasize it: "Lateralised dopaminergic feedback from the Lateral Accessory Lobes – a pre-motor area receiving path integration outputs – to specific compartments of

the MBs could represent an ideal candidate to orchestrate such a categorisation of left/right memories⁷⁰. Indeed, the insect brain receive massive lateralised signal from motor feedback^{85,86}, including such LAL to MB neurons (see for instance⁸⁷)."

49: orientations instead of orientation

65: Cartwright and Collett (1983) suggest that landmark navigation might depend on the allocentric reference frame. This should be cited.

82: remove extra "ants"

Answer: All done

Thank you very much for your time and helpful comments.

Dear editor,

We are grateful to the reviewers for their second round of feedbacks. We have incorporated all their suggestions into the revised manuscript and figures. Below our detail answers. With best wishes,

Antoine

Reviewer #1 (Remarks to the Author):

Thank you for addressing the comments in our previous review, we are certainly satisfied with the changes and clarifications provided and raise only a few minor comments:

L45-48: These lines are a helpful addition but they need a little bit of clarification/elaboration in my view.

Is the sentence beginning 'however', and in particular 'short-term mapping', referring to path integration as calculated with terrestrial cues? I think it could help to relate this to what I think you mean by 'path integration' in the next sentence (i.e path integration in general).

With respect to 'different surroundings', perhaps clarifying that you mean somewhere distinct from their collection point but still familiar e.g 'distinct yet familiar surroundings'?

Response: thanks, it is now clarified (Line 43 – 49). Explanation about path integration is provided in the next paragraph (Line 53-56).

Typos/ minor clarifications:

L44: 'without other source' -> without any other source/ without other sources/ without sources other than the terrestrial view (for more clarification).

L596: The part 'Turn ratio (integral of absolute angular velocity: (right - left) / (right + left)...' looks a little like the formula relates to the integral rather than the turn ratio, perhaps clarification with e.g. '(right - left) / (right + left), with right/left angles derived from the integral of absolute angular velocity'

L647: This should probably say 'green and orange arrows'.

L680, 689 & 691: check that colours in legend and figure are the same.

L681: just a slight rewording to 'yellow and orange marks, for which extent and thickness illustrate respectively the agent's recently sampled directions and familiarity signals received'

Response: all done

Supplementary materials:

Figure 1 is certainly a helpful addition, one minor point would be to consider the issue with the formula in line 7 (as discussed for the main text above).

L113: should say c)

Response: All done

Thanks again for your feedbacks.

Reviewer #3 (Remarks to the Author):

In the revised version of the manuscript the authors have addressed all my previous concerns. I have only three more minor points.

1. As far as I understand, a coordinating role of the LAL in locking turning signals to specific phases of an ant's motor oscillations has also been proposed in a recent review paper (Collett, T., Graham, P. and Heinze, S. (2025). The neuroethology of ant navigation. *Current Biology* 35, R110–R124.), maybe worth including in line 241 in the discussion.

Response: absolutely, done.

2. The dual role of delta7 cells, i.e. an ability to excite some neurons and inhibit other downstream neurons, was suggested via physiologically methods in Franconville, R., Beron, C. and Jayaraman, V. (2018). Building a functional connectome of the *Drosophila* central complex. *eLife* 7, e04577. This paper should be cited to justify the modeling choice shown in extended data figure 4.

Response: done. Indeed, it's worth highlighting this remarkable property.

3. The headings on top of the pages of the extended figures are out of sync with the figure legends (figure 3 is referred to as figure 2, and figure 4 is referred to as figure 3)

Response: corrected

Thanks again for your feedbacks.

Reviewer #4 (Remarks to the Author):

The authors have made changes that clarify some of the issues that we raised. These largely address the main concerns, but some additional edits would also be helpful. Specific points below:

In their first answer, the authors suggest that the views experienced by the ant are stored in a “fundamentally egocentric way” — i.e. that a static view cannot be used by the ants to extract an appropriate turning direction. To convince us they cite 3, 19, 20, 21, all of which propose that the parsimonious explanation for ant behavior is a view-matching strategy in which the ant only has access to a single familiarity value in each direction at one point along the route. They suggest that this is why they need stored views in multiple directions at each point along the route and the reason why ants need to scan left or right to perform route navigation. This is consistent with the result that ants carrying their food backwards cannot recognize a scene along a route they usually take in the forward direction. They also point out that the ants experience multiple views at each point along the route providing lots of opportunities to store “left of goal” and “right of goal” views.

These are all fair points. However, the authors' argument still relies on parsimony and does not rule

out the suggested alternative. In other words, IF the ant can compare two distinct familiarity values from the left and right MB for a single gaze direction to arrive at an instantaneous signed turning signal, this would also be consistent with the experimental results cited. This is why we requested that the authors show some of the scenes viewed by the ants at a few locations. If that is not feasible, we would suggest that the authors include a single sentence, perhaps in General Discussion if not Results, stating that further experimentation is required to definitively rule out the possibility that the comparison of an instantaneous view with a single stored snapshot at each point along the route is used by insects to immediately estimate which direction to turn.

Response: Agreed, we added this in the discussion (L219 -223).

For Answer 2, the citation in the text needs to be to the authors' bioRxiv manuscript, not Aso et al., 2014 (which is what reference 69 seems to point to).

Response: Indeed. Both citations are now present and the sentence clarified (L 166)

For Answer 8, the authors agree that the sinusoidal curve in Fig 1c can be explained without the need for lateralized signals from the MB to the PFNs. They should update the text to include that possibility.

Response: Indeed. The result of Fig.1c are interpreted purely behaviourally, no mention of neural circuit here (L50-72). When later referring to neural circuits, the uncertainty regarding the origin of this signal is clearly and systematically conveyed (e.g. L14 'assumed to be encoded in the Mushroom Bodies^{3,7-9'} ; L 116 'presumably computed in the MB' - L127 'presumably computed in the MB')

For Answer 9, Extended data fig 1 only shows data for a familiar location, but it would be useful to also see it for an unfamiliar location, as we suggested.

Response: Done. We added these detailed figure to extended data fig. 1

With the few changes above, the manuscript would be appropriate for publication without further review.

Thanks again for your feedbacks.

Reviewer #5 (Remarks to the Author):
